# The Differential Clinical Impacts of Cachexia and Sarcopenia on the Prognosis of Advanced Pancreatic Cancer

**DOI:** 10.3390/cancers14133137

**Published:** 2022-06-26

**Authors:** Ya-Chin Hou, Chien-Yu Chen, Chien-Jui Huang, Chih-Jung Wang, Ying-Jui Chao, Nai-Jung Chiang, Hao-Chen Wang, Hui-Ling Tung, Hsiao-Chun Liu, Yan-Shen Shan

**Affiliations:** 1Institute of Clinical Medicine, College of Medicine, National Cheng Kung University, Tainan 704, Taiwan; yachi2016@yahoo.com.tw (Y.-C.H.); poemcage@gmail.com (C.-J.W.); esr2wang@gmail.com (H.-C.W.); 2Department of Clinical Medical Research, National Cheng Kung University Hospital, College of Medicine, National Cheng Kung University, Tainan 704, Taiwan; antony910912@gmail.com; 3Division of General Surgery, Department of Surgery, National Cheng Kung University Hospital, College of Medicine, National Cheng Kung University, Tainan 704, Taiwan; surgeon.zhao@gmail.com (Y.-J.C.); ritaiap.tw@gmail.com (H.-L.T.); 4School of Medicine, College of Medicine, National Cheng Kung University, Tainan 704, Taiwan; a0978337257@gmail.com; 5Division of Gastroenterology and Hepatology, Department of Internal Medicine, National Cheng Kung University Hospital, College of Medicine, National Cheng Kung University, Tainan 704, Taiwan; nelly91.huang@gmail.com; 6Department of Oncology, Taipei Veterans General Hospital, Taipei 112, Taiwan; njchiang@nhri.edu.tw; 7School of Medicine, National Yang Ming Chiao Tung University, Taipei 112, Taiwan; 8National Institute of Cancer Research, National Health Research Institutes, Tainan 704, Taiwan; 9College of Medicine, National Cheng Kung University, Tainan 704, Taiwan

**Keywords:** pancreatic cancer, advanced stage, cancer cachexia, sarcopenia, age, body mass index, hemoglobin, albumin

## Abstract

**Simple Summary:**

Pancreatic cancer (PC) is one of the most lethal malignancies across the world. More than 80% patients are diagnosed at an advanced stage with limited treatment options. PC has the highest frequency of developing cancer cachexia (CC)–sarcopenia (SC) syndrome, but there is no optimal efficient intervention for CC or SC targeting their complicated biological and irreversible processes. As a result, identifying the clinicopathological features and patient characteristics in each advanced PC patient with CC and/or SC is urgently needed to distinguish different wasting phenotypes or body composition and thus support precision medicine and achieve optimal outcomes. We performed a retrospective cohort study of 232 advanced PC patients to determine the differential clinical characteristics of CC and SC and the overlap of CC- or SC-related factors in each patient. The impacts of CC, SC, and their relevant factors on the outcomes of PC patients were also evaluated.

**Abstract:**

Pancreatic cancer (PC) has the highest frequency of developing cancer cachexia (CC)–sarcopenia (SC) syndrome, which negatively influences patients’ outcome, quality of life, and tolerance/response to treatments. However, the clinical impacts of CC, SC, and their associated factors on outcomes for advanced PC has yet to be fully investigated. A total of 232 patients were enrolled in this study for the retrospective review of their clinical information and the measurement of skeletal muscle areas at the third lumber vertebra by computed tomography scan to identify CC or SC. The association and concurrent occurrence of clinicopathological features in each patient, prevalence rates, and prognosis with the CC or SC were calculated. CC and SC were observed in 83.6% (*n* = 194) and 49.1% (*n* = 114) of PC patients, respectively. Low hemoglobin levels more often occurred in CC patients than in non-CC patients (*p* = 0.014). Older age (*p* = 0.000), female gender (*p* = 0.024), low body mass index (BMI) values (*p* = 0.004), low hemoglobin levels (*p* = 0.036), and low albumin levels (*p* = 0.001) were more often found in SC patients than in non-SC patients. Univariate and multivariate analyses showed that CC was an independent poor prognostic factor of overall survival (OS) and progression-free survival for all patients, the chemotherapy (C/T) subgroup, and the high BMI subgroup. Meanwhile, SC was an independent predictor of poor OS for the subgroups of C/T or high BMI but not for all patients. These findings reveal the clinical differences for CC and SC and provide useful information for predicting the prognosis of advanced PC patients and conducting personalized medicine.

## 1. Introduction

Pancreatic cancer (PC) is the seventh leading cause of cancer death in both sexes worldwide in 2020 [1]. PC incidence rates are almost equal to its mortality rates. The prognosis has improved but minimally over the past few decades [1,2]. At initial diagnosis, more than 80% of patients progress to either metastatic or locally advanced disease with limited treatment options [3]. Due to delayed diagnosis, PC has the highest frequency of developing cancer cachexia (CC)–sarcopenia (SC) syndrome, leading to dismal patient outcomes, reduced quality of life, and poor tolerance/response to treatments [4,5].

CC is a complex, multifactorial, and continuum condition characterized by substantial weight loss with specific loss of skeletal muscle and body fat [4], affecting around 80% of PC patients and responsible for 50% of all cancer deaths worldwide [4,5]. SC is defined as the progressive loss of muscle mass and function and is associated with general loss of body mass, which may also occur with obesity in a condition identified as sarcopenic obesity (SO) [6]. Up to 50% of PC patients have SC, while the prevalence is about 40% in other common cancers [7,8]. A few studies and our previous work have found that both CC and SC correlated with poor prognosis in patients with malignancies, particularly PC [9,10,11]; however, the correlation usually depends on cancer types or stages.

To date, there is no optimal efficient intervention for CC or SC against their complicated biological and irreversible processes [6,12]. Therefore, identifying the clinicopathological features and patient characteristics in advanced PC patients with CC or SC is urgently needed in order to better select patients who can benefit from different and/or personalized approaches and use these therapeutic strategies at the optimal time point. Furthermore, to treat CC or SC optimally, any interventions should be initiated as early as possible to maximize the therapeutic efficacy and slow the development and progression of CC and SC [6,13], highlighting that early screening and careful monitoring are imperative for appropriate interventions.

Several patient-, tumor-, and treatment-related factors influencing prevalence of CC or SC have been reported [14,15]; however, the most suitable ones for CC or SC prediction in advanced PC were not suggested. Their concurrent occurrence might also have implications for treatment or outcomes. Accordingly, the present study aimed to identify the differential clinical characteristics and the overlap of associated factors for CC and SC. We also attempted to clarify the impacts of CC, SC, and their relevant factors on the outcomes of advanced PC patients.

## 2. Materials and Methods

### 2.1. Clinical Informatics Collection and CC or SC Definition

The study was approved by the Institutional Review Board (IRB) of the National Cheng Kung University Hospital (NCKUH; Tainan, Taiwan) (IRB number: B-ER-106-240 and B-ER-110-420). A total of 232 advanced PC patients were enrolled in this study, and the clinical information regarding patient demographics, clinicopathological parameters, and biochemical features from all patients between 2011 and 2021 were retrospectively reviewed and collected using electronic medical records (EMRs) under an IRB-approved protocol at NCKUH. Overall survival (OS) was measured as the period between initial diagnosis and death from any cause or the last visit to the outpatient center. Progression-free survival (PFS) was defined as the duration from the time of initial diagnosis to disease progression or death. The hemoglobin and albumin levels in the blood at diagnosis were determined using colorimetric assays on an automatic analyzer (Beckman, Fullerton, CA, USA) and performed in the Department of Pathology at NCKUH. The defined criterion for CC was weight loss greater than 5% over past 6 months or weight loss greater than 2% in individuals already showing depletion according to current bodyweight and height [body-mass index (BMI) < 20 kg/m^2^] or skeletal muscle mass (SC) based on a previous report [16]. According to the international consensus [17], total psoas area (TPA) index of <385 mm^2^/m^2^ in women and <545 mm^2^/m^2^ in men was used as the diagnostic criterion to define SC. The algorithm of the study cohort assembly is presented in Figure 1.

### 2.2. Computed Tomography (CT) Image Analysis and TPA Index Calculation

Abdominal CT scans were performed at 3-month intervals for diagnostic purposes and retrospectively evaluated on standard desktop computer screens through the Pictures Archiving and Communication System (PACS, version 3.0.10.2 (BM2), INFINITT Healthcare, Seoul, Korea). As described previously [9], TPA was estimated as a cross-sectional area of psoas muscle (mm^2^) based on Hounsfield unit (HU) thresholds ranging from −30 to +150 HU in a single image at the third lumbar vertebrae (L3) level using CT scan data. The TPA index was calculated using the formula: TPA divided by the square of the patient’s height (mm^2^/m^2^).

### 2.3. Overlap Assessment of CC, SC, and Their Associated Factors

The data set of PC patients with CC presence, SC presence, older age, high BMI (>22 kg/m^2^), low hemoglobin (≤12 g/dL), and low albumin (≤3.5 g/dL) was converted to the InteractiVenn (http://www.interactivenn.net/) (accessed on 1 December 2021) format and thus produced the Venn diagram as described in the previous study [18].

### 2.4. Statistical Analysis

All statistical data were generated using SPSS 17.0 software (SPSS, Chicago, IL, USA) and Prism 5.0 software (GraphPad Software, La Jolla, CA, USA). The chi-square test was used to detect differences in characteristics between patients with and without CC or SC. In case of a significant result, univariate and multivariate logistic regression analyses were applied to evaluate the effect and expressed as an odds ratio (OR) with a 95% confidence interval (CI). The survival rates of all patients and subgroups were calculated using the Kaplan–Meier method, and differences between curves were evaluated using the log-rank test. Univariate and multivariate analyses with a Cox proportional hazard model were performed to assess the independent prognostic factors, and the results are shown as a hazard ratio (HR) with a 95% CI. For the study, *p* values less than 0.05 were considered statistically significant.

## 3. Results

### 3.1. Survival Rates in Advanced PC Patients According to Clinical and Pathological Characteristics

The clinicopathologic features of the 232 patients are summarized in Table 1. There were 59.9% with younger age and 40.1% with older age. Among all patients, 64.2% were male and 35.8% were female, while 25.9% were in stage III and 74.1% were in stage IV. In all tumors, 40.1% were in the head, neck, or uncinate process, and 59.9% were in the body or tail of the pancreas. There were 6.5%, 36.6%, and 18.5% of cases with well-, moderately, or poorly differentiated PC, respectively. For treatment type, 10.3% of patients underwent conversion surgery with pre- or post-operative adjuvant chemotherapy (CS + adj subgroup), 74.1% received chemotherapy alone (C/T subgroup), and 15.5% received combination chemotherapy and local radiotherapy (C/T + local R/T subgroup). There were 69.0% with high carbohydrate antigen 19-9 (CA19-9) levels (>100 U/mL), 50.0% with low BMI values (≤22 kg/m^2^), 38.8% with low hemoglobin levels (≤12 g/dL), and 29.7% with low albumin levels (≤3.5 g/dL). Of the above-mentioned parameters, tumor stage (*p* = 0.000); treatment type (*p* = 0.000); BMI (*p* = 0.006); and levels of CA19-9 (*p* = 0.019), hemoglobin (*p* = 0.002), and albumin (*p* = 0.000) were associated with OS, while tumor stage (*p* = 0.050) and treatment type (*p* = 0.039) were related to PFS.

### 3.2. Comparison of Survival Outcomes among Advanced PC Patients, Patients with C/T Treatment or High BMI Values Stratified by CC or SC Status

The OS and PFS curves for all PC patients with or without CC and SC are shown in Figure 2. Patients in the non-CC group had significantly better OS and PFS than those in the CC group. The median OS (mOS) and PFS time (mPFS) of the patients without CC was 17.050 (95% CI: 10.504–23.596) and 8.210 (95% CI: 6.578–9.842) months, whereas mOS and mPFS of patients with CC were 9.430 (95% CI: 8.006–10.854) and 5.950 (95% CI: 5.133–6.767) months, respectively (*p* = 0.005, Figure 2A; *p* = 0.036, Figure 2B). No significant differences were found in the OS or PFS between patients with and without SC (*p* = 0.107, Figure 2C; *p* = 0.530, Figure 2D).

Because chemotherapy is the main treatment option for advanced PC but is also involved in CC development and progression [19,20] and the worst survival outcome was seen in the C/T subgroup when classifying patients by treatment type (Table 1), we sub-evaluated the importance of CC and SC on the prognosis of patients within the C/T subgroup (*n* = 172). OS and PFS for patients receiving chemotherapy treatment were significantly shorter in the CC group than in the non-CC group [mOS: 8.250 versus (vs.) 14.550 months, *p* = 0.003, Figure 3A; mPFS: 5.680 vs. 8.670 months, *p* = 0.005, Figure 3B, respectively]. There was a statistically significant difference in OS but not in PFS between patients with and without SC (mOS: 6.540 vs. 10.710 months, *p* = 0.009, Figure 3C; mPFS: 5.880 vs. 6.770 months, *p* = 0.289, Figure 3D, respectively).

Given that SC is an independent prognostic factor in resected PC patients with a BMI ≥ 22 [21], the mean BMI was 22, and BMI > 22 was related to poor OS (Table 1) in all subjects enrolled in this study, the patients with BMI greater than 22 (*n* = 116) were stratified by CC and SC. In this subgroup, CC patients had worse OS and PFS compared with non-CC patients (mOS: 8.250 vs. 15.240 months, *p* = 0.024, Figure 4A; mPFS: 5.780 vs. 10.940 months, *p* = 0.025, Figure 4B, respectively). OS but not PFS was also worse in SC patients than in non-SC patients (mOS: 6.240 vs. 10.610 months, *p* = 0.048, Figure 4C; mPFS: 5.780 vs. 6.010 months, *p* = 0.530, Figure 4D, respectively).

### 3.3. Comparison of Clinical Characteristics and Significance Levels Based on CC and SC

A total of 232 advanced PC patients were included in this study, among whom CC was present in 83.6% (*n* = 194) and SC in 49.1% (*n* = 114) (Figure 1). As shown in Table 2, CC patients more often had low hemoglobin levels than did non-CC patients (*p* = 0.014). Older age (*p* = 0.000), female gender (*p* = 0.024), and low BMI (*p* = 0.004), hemoglobin (*p* = 0.036), and albumin (*p* = 0.001) were more often observed in patients with SC than in patients without SC. Logistic regression analysis was further performed to assess the significance of these factors in predicting patients with CC or SC (Table 3). Both univariate and multivariate analyses revealed that low hemoglobin levels were a significant independent variable in CC prediction (OR: 2.746, 95% CI: 1.197–6.298, *p* = 0.017; OR: 2.718, 95% CI: 1.156–6.391, *p* = 0.022, respectively). According to univariate analysis, older age (OR: 2.862, 95% CI: 1.660–4.935, *p* = 0.000), female gender (OR: 1.862, 95% CI: 1.081–3.210, *p* = 0.025), low BMI (OR: 2.156, 95% CI: 1.276–3.642, *p* = 0.004), low levels of hemoglobin (OR: 1.766, 95% CI: 1.035–3.011, *p* = 0.037) and low levels of albumin (OR: 2.794, 95% CI: 1.550–5.038, *p* = 0.001) were related to SC. In the multivariate analysis, patients with older age (OR: 2.745, 95% CI: 1.534–4.913, *p* = 0.001), low BMI (OR: 2.492, 95% CI: 1.409–4.405, *p* = 0.002), and low albumin levels (OR: 2.648, 95% CI: 1.401–5.004, *p* = 0.003) were at higher risk of developing SC.

### 3.4. Prognostic Significance of Clinical Factors and CC or SC Presence

The prognostic importance of factors influencing OS and PFS in the Cox regression model are listed in Table 4. The univariate analysis showed that disease stage (HR: 1.837, 95% CI: 1.321–2.556, *p* = 0.000), treatment strategies (HR: 3.685, 95% CI: 1.941–6.996, *p* = 0.000 for C/T subgroup/CS + adj subgroup; HR: 2.702, 95% CI: 1.334–5.472, *p* = 0.006 for C/T + local R/T subgroup/CS + adj subgroup, respectively), high CA19-9 levels (HR: 1.442, 95% CI: 1.059–1.963, *p* = 0.020), high BMI (HR: 1.478, 95% CI: 1.118–1.956, *p* = 0.006), low hemoglobin levels (HR: 1.555, 95% CI: 1.170–2.065, *p* = 0.002), and low albumin levels (HR: 2.508, 95% CI: 1.855–3.390, *p* = 0.000) were independent prognostic factors significantly associated with poor OS, while treatment strategies (HR: 1.877, 95% CI: 1.104–3.193, *p* = 0.020 for C/T subgroup/CS + adj subgroup; HR: 2.076, 95% CI: 1.138–3.785, *p* = 0.017 for C/T + local R/T subgroup/CS + adj subgroup, respectively) were independent risk factors related to poor PFS. The multivariate analysis indicated that disease stage (HR: 1.744, 95% CI: 1.240–2.453, *p* = 0.001), treatment type (HR: 3.214, 95% CI: 1.678–6.157, *p* = 0.000 for C/T subgroup/CS + adj subgroup), BMI (HR: 1.656, 95% CI: 1.240–2.211, *p* = 0.001), and albumin levels (HR: 2.330, 95% CI: 1.693–3.207, *p* = 0.000) were independent prognostic factors for OS, while disease stage (HR: 1.465, 95% CI: 1.005–2.135, *p* = 0.047) was an independent risk factor for PFS.

The prognostic values of CC and SC status in all patients and in the C/T or high BMI subgroups were also measured with a Cox proportional hazard model. The univariate and multivariate analyses showed that the presence of CC was an independent predictor of poor OS for all patients, the C/T subgroup, and the high BMI subgroup, with statistical significance (*p* = 0.006 and 0.000; *p* = 0.004 and 0.003; and *p* = 0.026 and 0.019, respectively). Meanwhile, both univariate and multivariate analyses revealed that the presence of SC was an independent predictor of poor OS for the C/T subgroup and the high BMI subgroup but not all patients (*p* = 0.010 and 0.008; *p* = 0.049 and 0.033; *p* = 0.108 and 0.193, respectively). Similarly, CC presence was also an independent poor prognostic factor of PFS for all patients, the C/T subgroup, and the high BMI subgroup determined by the univariate and multivariate analyses (*p* = 0.038 and 0.017; *p* = 0.007 and 0.008; *p* = 0.029 and 0.031, respectively); however, no significant differences in PFS between the presence and the absence of SC were observed in all patients or in these subgroups (*p* = 0.531 and 0.583; *p* = 0.290 and 0.382; *p* = 0.373 and 0.430, respectively).

### 3.5. Overlaps among CC, SC, and Their Associated Factors in Advanced PC

Based on the data in Table 2 and Table 3, the clinical parameters including older age and low BMI, hemoglobin, and albumin were correlated with CC or SC presence. Of note, high BMI was an independent predictor of shorter OS (Table 1). We therefore determined the overlaps of CC presence, SC presence, older age, high BMI, low hemoglobin, and low albumin to clarify the impacts of these factors either alone or combination on advanced PC. The overlaps among CC, SC, and their associated factors is displayed in a Venn diagram and presented in Figure 5. There was 50.5% (*n* = 98), 50.0% (*n* = 97), 42.3% (*n* = 82), 40.7% (*n* = 79), and 30.9% (*n* = 60) of CC patients (*n* = 194) also with SC, high BMI, low hemoglobin levels, older age, and low albumin levels, respectively. Among these patients, 30.4% (*n* = 59) had high BMI and with (*n* = 37)/without (*n* = 22) other factors, 5.2% (*n* = 10) with either or both albumin and hemoglobin in low levels, 4.6% (*n* = 9) with older age and with/without others, and 9.3% (*n* = 18) without additional factors. In the SC group (*n* = 114), patients affected by CC, older age, low hemoglobin levels, and high BMI as well as low albumin levels or not were 86.0% (*n* = 98), 52.6% (*n* = 60), 45.6% (*n* = 52), 40.4% (*n* = 46), 40.4% (*n* = 46), and 1.8% (*n* = 2), respectively. In the CC and SC co-presence group, older age was most commonly found in 52 of 98 cases (53.1%), followed by low hemoglobin levels (48.0%, *n* = 47) and low albumin levels (39.8%, *n* = 39) as well as high BMI (38.8%, *n* = 38). There were 52 cases (53.1%) with older age and with (*n* = 40)/without (*n* = 12) other factors, 18 cases (18.4%) with high BMI and with/without others, 18 cases (18.4%) with either or both albumin and hemoglobin in low levels, and 10 cases (10.2%) without additional factors.

## 4. Discussion

Advanced cancer, especially PC, has a high risk of developing CC and SC, which can impair quality of life, tolerance/response to cancer treatments, and prognosis; prolong hospitalizations; and increase the use of health care resources [4,5,6]. Unfortunately, to date, no approved drug therapies or efficient medical interventions can be used to improve these intricate, multifactorial, and irreversible conditions [6,12,13]. Identifying useful prognostic factors and individual patients’ characteristics are critical for instituting preventive intervention strategies or appropriate managements for advanced PC patients with CC or SC; however, little attention has so far been paid to this field. Here, we identified the differences in prognostic significance, clinical characteristics, and relevant factors for CC and SC in advanced PC patients. CC presence was an independent unfavorable prognostic factor of OS and PFS for all patients and the C/T and high BMI subgroups, whereas SC presence was an independent predictor only in poor OS for the C/T and high BMI subgroups but not all patients. Low hemoglobin level was associated with CC while older age, female gender, and low BMI, hemoglobin levels, and albumin levels were related to SC. When covering CC, SC, and their relevant factors simultaneously in patients with either or both CC and SC, considerable overlap was found: SC and/or high BMI in CC patients, CC in SC patients, and aging and/or low hemoglobin levels in CC and SC patients.

Due to the differences in the patient cohorts, assessment methodology, and diagnostic criteria used [22,23], the prevalence rates of CC and SC among PC patients are heterogeneous, ranging from 21.3 to 80% [4,5,22,24] and 28 to 89% [7,8,24], respectively. The largest cohort study including 977 PC patients using the same diagnostic criterion showed that 63% of patients were identified with CC, which was independently associated with worse OS but not related to all stages of disease, BMI classes, or receipt of chemotherapy. Among CC patients, the prevalence was higher in patients with unresectable disease compared with those receiving surgical resection (82.2% vs. 17.8%) [25]. In the present study, we enrolled mainly patients with unresectable PC and observed similar percentages of CC patients in all patients (83.6%), the C/T subgroup (84.3%), and the high BMI subgroup (83.6%). Of note, our previous work found that the prevalence rates of CC were 75.3%, 64.8%, and 92.7% in all 146 PC cases, the resected subgroup, and the locally advanced subgroup, respectively [9]. The finding obtained in the present work was also consistent with our previous report [9] showing that CC patients had significantly shorter OS and DFS than those without CC. Hence, CC is strongly related to advanced PC and progresses throughout the disease course.

Although it has been recommended that SC definitions include loss of muscle mass, strength, and physical performance [26], measurements of skeletal muscle cross-sectional area at the L3 level derived from CT images are commonly used for SC assessment in numerous studies and have become standard [8]. Using this approach, the SC prevalence rates were 47.8% (*n* = 69), 49.1% (*n* = 53), 51.2% (*n* = 82), 63.4% (*n* = 41), 64.6% (*n* = 215), 73.4% (*n* = 94), 59.3% (*n* = 251), and 55.9% (*n* = 111) in PC patients with inoperable locally advanced [27,28,29,30,31,32] or metastatic disease [33] or entering a palliative therapy program [34], respectively. Some [29,31], but not all [27,28,30,32,33,34], studies have found that SC was significantly associated with unfavorable OS and severe chemotherapy toxicity [33]. SC represents a negative prognostic factor in PC patients receiving FOLFIRINOX [29] or palliative first-line gemcitabine-based chemotherapy [33] or undergoing endoscopic ultrasound celiac plexus neurolysis [31]. The effect on survival and treatment response was obvious in SO. SC patients with overweight or obesity had worse OS when compared to others [28,30,34], while SO predicted severe hematologic toxicity in PC patients who received FOLFIRINOX [29]. In agreement with these studies, our results showed that SC was observed in 49.1%, 52.3%, and 39.7% of all patients, the C/T subgroup, and the high BMI subgroup, respectively. SC presence as an independent predictor in poor OS within the subgroups but not all patients.

Like a vicious cycle, not only does anticancer therapy potentiate CC and SC, but CC and SC also increase the toxicities of chemotherapy or chemoradiation and reduce the surgical resectability or benefits from a subsequent resection in patients with locally advanced PC treated with neoadjuvant therapy [35,36,37]. Cancer patients have been shown to have a higher rate of muscle loss than healthy adults over the age of 40 years, generally losing muscle mass at a rate of 1–1.4% per year [38]. Loss of muscle in PC patients undergoing chemotherapy has been examined at a rate of 2.9% every 100 days [39]. The rate is comparable with the observation of advanced PC patients receiving palliative care, who lose muscle at a rate of 3.1% every 100 days [34]. In a foregut cancer cohort study that included the esophagus (*n* = 99), stomach (*n* = 39), pancreas (*n* = 55), liver (*n* = 4), and bile ducts (*n* = 28), patients treated with neoadjuvant chemotherapy experienced greater losses in muscle compared with those treated with palliative chemotherapy (7.3% vs. 2.8% every 100 days). Among patients treated with neoadjuvant chemotherapy, the non-responders lost significantly more muscle than responders (8.7% vs. 4.0% every 100 days) [39]. Similarly, the highest prevalence of SC was found in the C/T subgroup than in the others in our study. Since SC is a mere component of CC, deliberating SC before advanced PC patients entering anticancer therapy or support care programs can improve patient care and outcome.

CC has commonly been considered an energy-wasting syndrome, in which tumor-driven systemic inflammation induces metabolic alterations and thus promotes tumor growth and expansion. The balance of energy intake or expenditure depends on patient response to tumor progression [5]. SC is invariably characterized by muscle atrophy that is a dynamic process in humans related to imbalances between muscle protein catabolism and anabolism and occurs with aging, malnutrition, immobility, inflammatory disease, and cancer [6]. Because CC and SC represent extremely complex biological processes and cannot be fully reversed by conventional nutritional support [16,35,40], there is some evidence that multimodal therapies may have a better chance of attenuating the progression of these conditions [4,41]. To really address the issue, our study addressed the relationships among clinicopathological features, patient characteristics, and their concurrent occurrence. We observed that CC presence was correlated with low hemoglobin levels, while SC presence was related to aging, female gender, low BMI, low hemoglobin, and low albumin, supported by previous studies that implicated older age [21], female gender [27], and low BMI [21,27,29,34] in PC patients with SC.

Although there are no specific laboratory markers for pancreatic CC, hemoglobin and albumin have been reported to be involved in gastrointestinal, gynecological, hematopoietic, lung cancer [15], and PC [37]. Hemoglobin and albumin levels were associated with a statistically significant reduction in survival of CC patients [42,43] and were also significantly decreased in locally advanced PC patients after neoadjuvant chemoradiation treatment [37]. Of note, our previous study showed that elevated CA19-9 levels (>100 U/mL) were associated with shorter OS in patients with locally advanced PC compared with those with resected disease [9]. Consistent results were observed in the survival analysis in this study and showed worse OS for patients who had CA19-9 levels of more than 100 U/mL. These results suggest that a CA19-9 level of >100U/mL acts as an indicator of disease progression. The cause of low hemoglobin and albumin levels in cancer patients is likely multifactorial. They are affected by radiotherapy- or chemotherapy-related toxicities [44] and the nutritional and metabolic alterations occurring during the cachectic process. This process involves abnormalities of the multi-organ functions and mediates systemic inflammation, resulting in anorexia (nutritional deprivation), reduced albumin synthesis, or cancer-related anemia [5,45]. Moreover, low hemoglobin and albumin levels were at risk of poor nutritional status and contributed to the development of sarcopenia [46,47]. These findings reveal that these routinely available clinical markers can serve as indicators of CC or SC in PC.

Even in a highly homogeneous population of PC patients with equivalent stage, diagnosis, and treatment, three distinct CC phenotypes including muscle and fat wasting, fat-only wasting, and no wasting could be found [28]. We further investigated the overlap of CC or SC-related factors in each patient within this homogenous cohort and identified that these factors partially overlapped and were interrelated. Half of the CC patients had SC and/or high BMI and more than 80% of SC patients were also cachectic. CC and SC occurred concurrently in 42.2% of 232 PC patients, and these patients were affected by aging and/or low hemoglobin levels. These data suggest that the differences in wasting pattern or body composition are driven by host or tumor heterogeneity, highlight the need to depict patients’ personalized characteristics, and warrant further investigation to early recognition and treatment of the nutritional and metabolic alterations occurring during disease progression and slow the wasting process.

## 5. Conclusions

CC represents an independent unfavorable prognostic factor of OS and PFS for all patients and subgroups, whereas SC can be considered a negative prognostic factor for OS in advanced PC patients within C/T or high BMI subgroups. CC or SC-related factors including older age, female gender and low BMI, low hemoglobin, and low albumin partially overlap and are interrelated. Future studies are needed to comprehensively understand the effects and overlap of relevant factors in PC patients with CC or SC to distinguish different wasting phenotypes or body compositions and thus support precision medicine and achieve optimal outcomes.

## Figures and Tables

**Figure 1 cancers-14-03137-f001:**
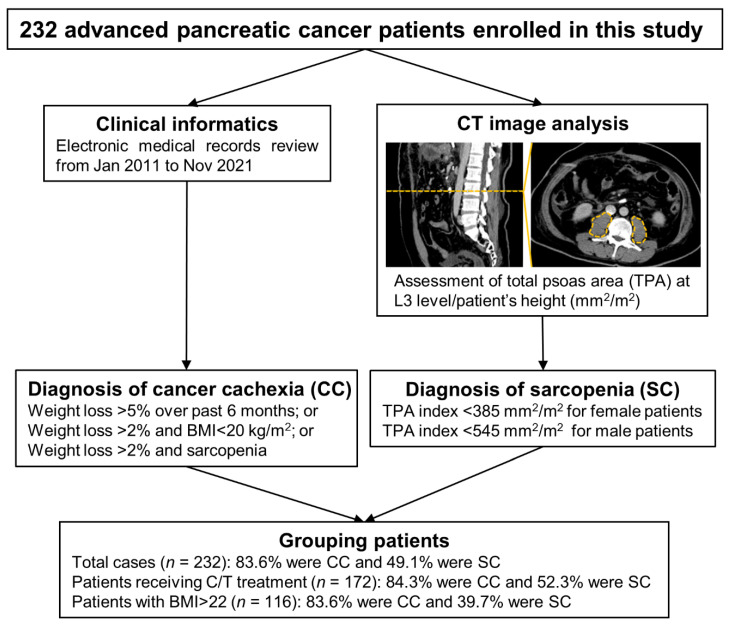
Study design flowchart illustrating the process of participant recruitment, data collection, and analysis throughout this study. CT, computed tomography; TPA, total psoas area; L3, the third lumbar vertebrae; CC, cancer cachexia; BMI, body mass index; SC, sarcopenia: C/T, chemotherapy.

**Figure 2 cancers-14-03137-f002:**
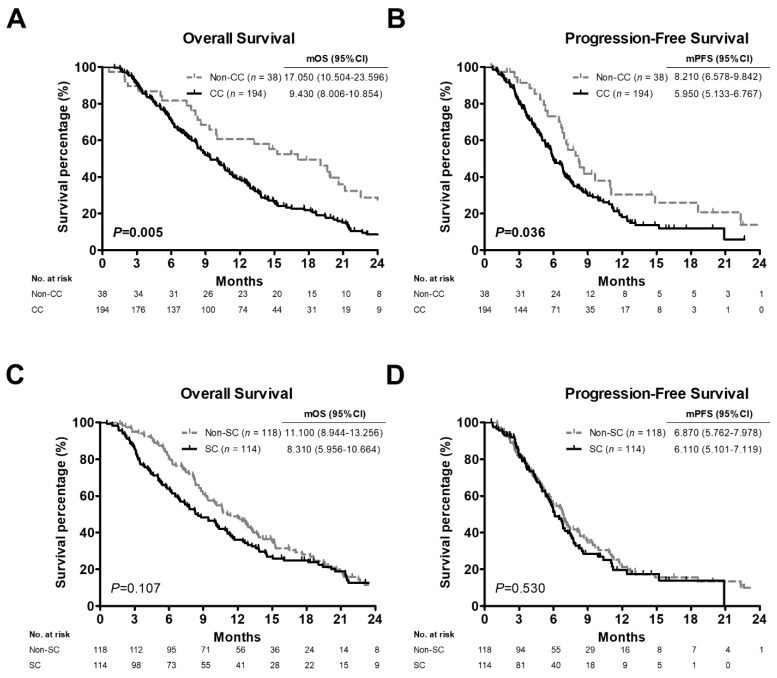
Survival curves of 232 advanced pancreatic cancer patients based on cancer cachexia or sarcopenia status generated by Kaplan-Meier analyses. (**A**,**C**) Overall survival curves of patients with cancer cachexia (**A**) or sarcopenia (**C**). (**B**,**D**) Progression-free survival curves of patients with cancer cachexia (**B**) or sarcopenia (**D**). *p* values determined using the log-rank test. CC, positive cancer cachexia status; Non-CC, negative cancer cachexia status; SC, positive sarcopenia status; Non-SC, negative sarcopenia status; mOS, median overall survival; mPFS, median progression-free survival; CI, confidence interval.

**Figure 3 cancers-14-03137-f003:**
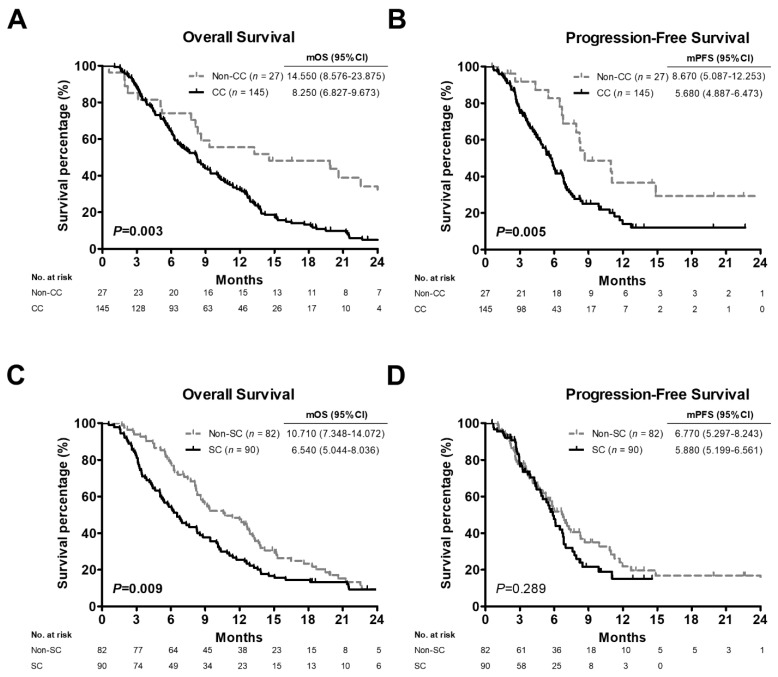
Kaplan-Meier estimate of survival rate in relation to cancer cachexia or sarcopenia status among 172 pancreatic cancer patients receiving chemotherapy treatment. (**A**,**C**) Overall survival curves of patients with cancer cachexia (**A**) or sarcopenia (**C**). (**B**,**D**) Progression-free survival curves of patients with cancer cachexia (**B**) or sarcopenia (**D**). *p* values determined using the log-rank test. CC, positive cancer cachexia status; Non-CC, negative cancer cachexia status; SC, positive sarcopenia status; Non-SC, negative sarcopenia status; mOS, median overall survival; mPFS, median progression-free survival; CI, confidence interval.

**Figure 4 cancers-14-03137-f004:**
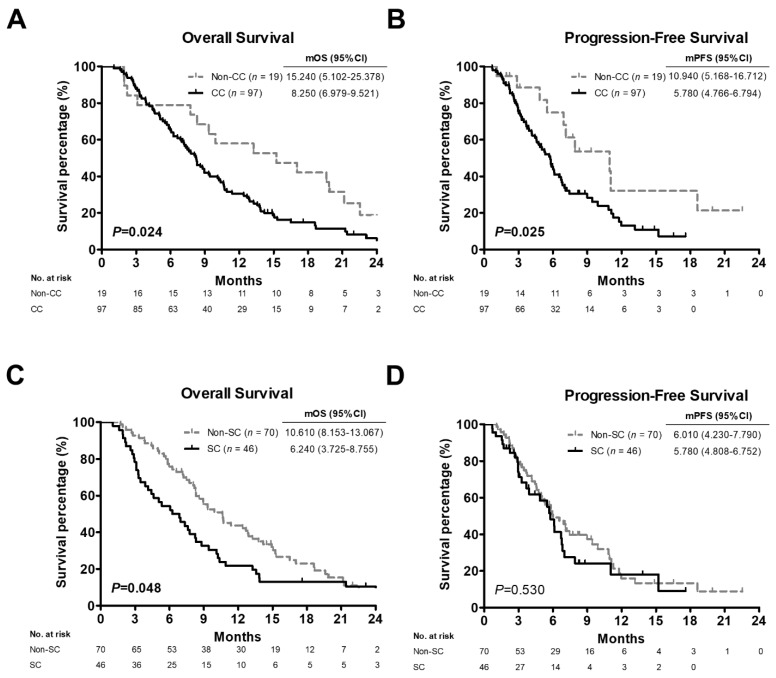
Kaplan-Meier analysis plotting the survival curves of 116 pancreatic cancer patients with BMI > 22 according to cancer cachexia or sarcopenia status. (**A**,**C**) Overall survival curves of patients with BMI > 22 stratified by cancer cachexia (**A**) or sarcopenia (**C**). (**B**,**D**) Progression-free survival curves of patients with BMI > 22 stratified by cancer cachexia (**B**) or sarcopenia (**D**). *p* values determined using the log-rank test. BMI, body mass index; CC, positive cancer cachexia status; Non-CC, negative cancer cachexia status; SC, positive sarcopenia status; Non-SC, negative sarcopenia status; mOS, median overall survival; mPFS, median progression-free survival; CI, confidence interval.

**Figure 5 cancers-14-03137-f005:**
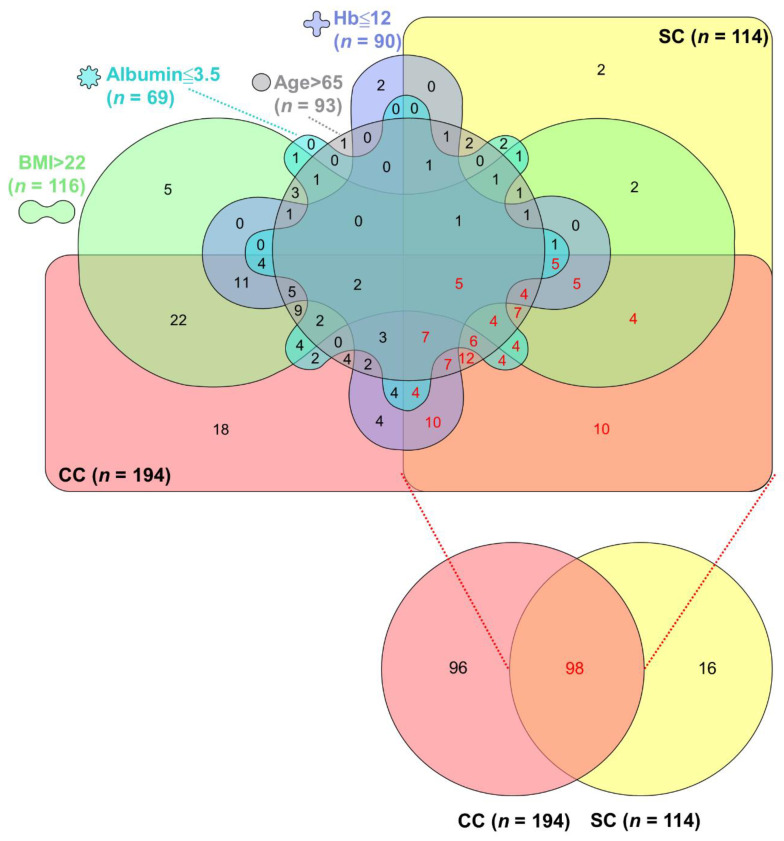
Overlap of six parameters including cancer cachexia presence, sarcopenia presence, older age, high BMI values, and low levels of hemoglobin or albumin in pancreatic cancer patients. The unique overlapping among positive cancer cachexia status (red), positive sarcopenia status (yellow), age > 65 years (gray), BMI > 22 kg/m^2^ (green), hemoglobin ≤ 12 g/dL (purple) and albumin ≤ 3.5 g/dL (blue) is illustrated in a Venn diagram. Concurrent CC and SC is shown in orange. The number in each section represents the number of cases, of which the number labeled in red indicates the cases categorized in the CC and SC co-presence group. CC, cancer cachexia; SC, sarcopenia; BMI, body mass index; Hb, hemoglobin.

**Table 1 cancers-14-03137-t001:** Clinical factors affecting overall survival and progression-free survival in patients with advanced pancreatic cancer.

Variable	Patients (%)	Overall Survival		Progression-Free Survival	
		Median (Months)	*p* Value	Median (Months)	*p* Value
Age					
≤65 years	139 (59.9)	10.580	0.114	6.110	0.254
>65 years	93 (40.1)	8.310		7.000	
Gender					
Male	149 (64.2)	10.220	0.853	6.930	0.147
Female	83 (35.8)	10.090		5.880	
Stage					
III	60 (25.9)	14.920	**0.000**	8.480	**0.050**
IV	172 (74.1)	8.610		6.470	
Tumor location					
Head/neck/uncinate process	93 (40.1)	10.150	0.880	6.830	0.774
Body/tail	139 (59.9)	10.120		6.110	
Tumor grade					
Well diff.	15 (6.5)	11.990	0.426	6.540	0.968
Moderately diff.	85 (36.6)	11.070		6.800	
Poorly diff.	43 (18.5)	8.110		6.670	
Unknown	89 (38.4)	10.220		6.700	
Treatment					
CS + adj	24 (10.3)	25.590	**0.000**	9.430	**0.039**
C/T	172 (74.1)	8.310		6.080	
C/T + local R/T	36 (15.5)	11.560		5.950	
CA19-9					
≤100 U/mL	72 (31.0)	12.160	**0.019**	6.970	0.067
>100 U/ml	160 (69.0)	9.170		6.670	
BMI					
≤22 kg/m^2^	116 (50.0)	11.700	**0.006**	6.900	0.181
>22 kg/m^2^	116 (50.0)	8.310		5.880	
Hemoglobin					
≤12 g/dL	90 (38.8)	8.280	**0.002**	5.950	0.258
>12 g/dL	142 (61.2)	12.160		6.930	
Albumin					
≤3.5 g/dL	69 (29.7)	5.720	**0.000**	7.260	0.859
>3.5 g/dL	163 (70.3)	12.910		6.540	

Abbreviations: diff., differentiation; CS, conversion surgery; adj, adjuvant chemotherapy; C/T, chemotherapy; R/T, radiotherapy; CA19-9, carbohydrate antigen 19-9; BMI, body mass index. The bold value indicates *p* < 0.05.

**Table 2 cancers-14-03137-t002:** Comparison of patient characteristics based on cancer cachexia and sarcopenia.

Variable	Cancer Cachexia (%)			Sarcopenia (%)		
	No (*n* = 38)	Yes (*n* = 194)	*p* Value	No (*n* = 118)	Yes (*n* = 114)	*p* Value
Age						
≤65 years	24 (63.2)	115 (59.3)	0.655	85 (72.0)	54 (47.4)	**0.000**
>65 years	14 (36.8)	79 (40.7)		33 (28.0)	60 (52.6)	
Gender						
Male	25 (65.8)	124 (63.9)	0.826	84 (71.2)	65 (57.0)	**0.024**
Female	13 (34.2)	70 (36.1)		34 (28.8)	49 (43.0)	
Stage						
III	5 (13.2)	55 (28.4)	0.051	28 (23.7)	32 (28.1)	0.450
IV	33 (86.8)	139 (71.6)		90 (76.3)	82 (71.9)	
Tumor location						
Head/neck/uncinate process	15 (39.5)	78 (40.2)	0.933	44 (37.3)	49 (43.0)	0.376
Body/tail	23 (60.5)	116 (59.8)		74 (62.7)	65 (57.0)	
Tumor grade						
Well diff.	2 (5.3)	13 (6.7)	0.461	9 (7.6)	6 (5.3)	0.902
Moderately diff.	18 (47.4)	67 (34.5)		43 (36.4)	42 (36.8)	
Poorly diff.	7 (18.4)	36 (18.6)		22 (18.6)	21 (18.4)	
Unknown	11 (28.9)	78 (40.2)		44 (37.3)	45 (39.5)	
Treatment						
CS + adj	3 (7.9)	21 (10.8)	0.544	13 (11.0)	11 (9.6)	0.197
C/T	27 (71.1)	145 (74.7)		82 (69.5)	90 (78.9)	
C/T + local R/T	8 (21.1)	28 (14.4)		23 (19.5)	13 (11.4)	
CA19-9						
≤100 U/mL	13 (34.2)	59 (30.4)	0.644	37 (31.4)	35 (30.7)	0.914
>100 U/mL	25 (65.8)	135 (69.6)		81 (68.6)	79 (69.3)	
BMI						
≤22 kg/m^2^	19 (50.0)	97 (50.0)	1.000	48 (40.7)	68 (59.6)	**0.004**
>22 kg/m^2^	19 (50.0)	97 (50.0)		70 (59.3)	46 (40.4)	
Hemoglobin						
≤12 g/dL	8 (21.1)	82 (42.3)	**0.014**	38 (32.2)	52 (45.6)	**0.036**
>12 g/dL	30 (78.9)	112 (57.7)		80 (67.8)	62 (54.4)	
Albumin						
≤3.5 g/dL	9 (23.7)	60 (30.9)	0.372	23 (19.5)	46 (40.4)	**0.001**
>3.5 g/dL	29 (76.3)	134 (69.1)		95 (80.5)	68 (59.6)	

Abbreviations: diff., differentiation; CS, conversion surgery; adj, adjuvant chemotherapy; C/T, chemotherapy; R/T, radiotherapy; CA19-9, carbohydrate antigen 19-9; BMI, body mass index. The bold value indicates *p* < 0.05.

**Table 3 cancers-14-03137-t003:** Univariate and multivariate analysis of the investigated factors for predicting patients with cancer cachexia or sarcopenia.

Variable	Univariate Analysis		Multivariate Analysis	
	OR (95% CI)	*p* Value	OR (95% CI)	*p* Value
Factor influencing cancer cachexia				
Hemoglobin (≤12/>12 g/dL)	2.746 (1.197–6.298)	**0.017**	2.718 (1.156–6.391)	**0.022**
Factors influencing sarcopenia				
Age (>65/≤65 years)	2.862 (1.660–4.935)	**0.000**	2.745 (1.534–4.913)	**0.001**
Gender (female/male)	1.862 (1.081–3.210)	**0.025**	1.450 (0.797–2.637)	0.224
BMI (≤22/>22 kg/m^2^)	2.156 (1.276–3.642)	**0.004**	2.492 (1.409–4.405)	**0.002**
Hemoglobin (≤12/>12 g/dL)	1.766 (1.035–3.011)	**0.037**	1.396 (0.770–2.531)	0.271
Albumin (≤3.5/>3.5 g/dL)	2.794 (1.550–5.038)	**0.001**	2.648 (1.401–5.004)	**0.003**

Abbreviations: OR, odds ratio; CI, confidence interval; BMI, body mass index; CC, cancer cachexia; SC, sarcopenia. The bold value indicates *p* < 0.05.

**Table 4 cancers-14-03137-t004:** Univariate and multivariate analysis of the overall survival and progression-free survival in patients with advanced pancreatic cancer.

Variable	Univariate Analysis		Multivariate Analysis	
	HR (95% CI)	*p* Value	HR (95% CI)	*p* Value
Overall Survival				
Stage (IV/III)	1.837 (1.321–2.556)	**0.000**	1.744 (1.240–2.453)	**0.001**
Treatment (C/T/CS + adj)	3.685 (1.941–6.996)	**0.000**	3.214 (1.678–6.157)	**0.000**
Treatment (C/T + local R/T/CS + adj)	2.702 (1.334–5.472)	**0.006**	1.964 (0.958–4.025)	0.065
CA19-9 (>100/≤100 U/mL)	1.442 (1.059–1.963)	**0.020**	1.336 (0.971–1.839)	0.075
BMI (>22/≤22 kg/m^2^)	1.478 (1.118–1.956)	**0.006**	1.656 (1.240–2.211)	**0.001**
Hemoglobin (≤12/>12 g/dL)	1.555 (1.170–2.065)	**0.002**	1.305 (0.972–1.752)	0.076
Albumin (≤3.5/>3.5 g/dL)	2.508 (1.855–3.390)	**0.000**	2.330 (1.693–3.207)	**0.000**
CC (yes/no) in all patients	1.738 (1.176–2.568)	**0.006**	2.232 (1.474–3.379)	**0.000**
SC (yes/no) in all patients	1.256 (0.951–1.660)	0.108	1.210 (0.908–1.613)	0.193
CC (yes/no) in patients with C/T	2.001 (1.256–3.189)	**0.004**	2.037 (1.273–3.261)	**0.003**
SC (yes/no) in patients with C/T	1.517 (1.104–2.085)	**0.010**	1.533 (1.116–2.106)	**0.008**
CC (yes/no) in patients with BMI > 22	1.840 (1.075–3.151)	**0.026**	1.919 (1.112–3.313)	**0.019**
SC (yes/no) in patients with BMI > 22	1.488 (1.001–2.212)	**0.049**	1.537 (1.035–2.282)	**0.033**
Progression-Free Survival				
Stage (IV/III)	1.426 (0.997–2.040)	0.052	1.465 (1.005–2.135)	**0.047**
Treatment (C/T/CS + adj)	1.877 (1.104–3.193)	**0.020**	1.678 (0.969–2.906)	0.065
Treatment (C/T + local R/T/CS + adj)	2.076 (1.138–3.785)	**0.017**	1.795 (0.976–3.303)	0.060
CA19-9 (>100/≤100 U/mL)	1.388 (0.975–1.975)	0.069	1.356 (0.950–1.937)	0.094
CC (yes/no) in all patients	1.586 (1.026–2.451)	**0.038**	1.723 (1.103–2.692)	**0.017**
SC (yes/no) in all patients	1.107 (0.805–1.523)	0.531	1.094 (0.793–1.510)	0.583
CC (yes/no) in patients with C/T	2.212 (1.247–3.923)	**0.007**	2.183 (1.230–3.874)	**0.008**
SC (yes/no) in patients with C/T	1.233 (0.836–1.817)	0.290	1.189 (0.807–1.752)	0.382
CC (yes/no) in patients with BMI > 22	2.187 (1.084–4.409)	**0.029**	2.164 (1.073–4.363)	**0.031**
SC (yes /no) in patients with BMI > 22	1.237 (0.774–1.976)	0.373	1.208 (0.756–1.930)	0.430

Abbreviations: HR, hazard rate; CI, confidence interval; CS, conversion surgery; adj, adjuvant chemotherapy; C/T, chemotherapy; R/T, radiotherapy; CA19-9, carbohydrate antigen 19-9; BMI, body mass index; CC, cancer cachexia; SC, sarcopenia. Univariate and multivariate analyses for each risk factor were performed using the Cox regression model and show the HR and 95% CI data, and the bold value indicates *p* < 0.05.

## Data Availability

The data is contained within the article. The datasets generated and/or analyzed during the current study are not publicly available due to privacy issues.

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
