# Peer review of "The Differential Clinical Impacts of Cachexia and Sarcopenia on the Prognosis of Advanced Pancreatic Cancer"

_cancers, 2022, doi:10.3390/cancers14133137_

Round 1

Reviewer 1 Report

This paper is a cohort study analyzing the association between cancer cachexia and sarcopenia characteristics and prognosis in pancreatic cancer patients with the poorest prognosis. A large number of patients were included, and cancer cachexia was an independent poor prognostic factor for overall survival (OS) and progression-free survival. On the other hand, sarcopenia was an independent predictor of poor OS in the chemotherapy group and in the high BMI subgroup, but not in all patients, the authors concluded. These findings may be of interest to readers because they provide useful information in predicting prognosis in patients with advanced pancreatic cancer.

On the other hand, the following problems can be found. If the following items are improved, we will consider accepting the manuscript.

Major problems

1. In the Materials and Methods section, data analysis items and methods, especially analysis of overlaps among CC, SC, and their associated factors in advanced PC, are not sufficiently described.

2. In the Results section, there is a lot of repetition of data in the Tables, which is difficult to read.

Minor points
1. In Table 2, the heading of sarcopenia is not adequately described.
2. Table 3 would be more appropriately listed before Tables 1 and 2.

Author Response

Response to Reviewer 1 Comments

This paper is a cohort study analyzing the association between cancer cachexia and sarcopenia characteristics and prognosis in pancreatic cancer patients with the poorest prognosis. A large number of patients were included, and cancer cachexia was an independent poor prognostic factor for overall survival (OS) and progression-free survival. On the other hand, sarcopenia was an independent predictor of poor OS in the chemotherapy group and in the high BMI subgroup, but not in all patients, the authors concluded. These findings may be of interest to readers because they provide useful information in predicting prognosis in patients with advanced pancreatic cancer.

On the other hand, the following problems can be found. If the following items are improved, we will consider accepting the manuscript.

Reply to Reviewer 1

We thank the reviewer for his/her comments, which have helped to improve the paper and clarify the details of our study. Our point-by-point response is given below (labeled in red).

Major problems

  1. In the Materials and Methods section, data analysis items and methods, especially analysis of overlaps among CC, SC, and their associated factors in advanced PC, are not sufficiently described.

Response 1:

    Thank you very much for your comments. The description of detection procedure of hemoglobin and albumin in blood samples and the overlap assessment of CC, SC, and their associated factors was added in the Materials and Methods section. Please see the page 5 lines 17-19 and page 7 line 4-8.

  1. In the Results section, there is a lot of repetition of data in the Tables, which is difficult to read.

Response 2:

Thank you very much for your comments. We have deleted repetition of data in the Result section. Please see the page 16 line 26 and page 17 line 3 and 6.

Minor points

  1. In Table 2, the heading of sarcopenia is not adequately described.

Response 3:

Thank you very much for your suggestion. We revised Table 2 (now Table 3) and changed the heading of sarcopenis to “Factors influencing sarcopenia”. Please see the page 15.

  1. Table 3 would be more appropriately listed before Tables 1 and 2.

Response 4:

Thank you very much for your suggestion. We revised Table 3 (now Table 1, page 8), Table 1 (now Table 2, page 14), and Table 2 (now Table 3, page 15). Changes also made in the Result section. Please see the page 7 line 23 to page 8 line 12, page 9 line 5 to page 10 line 17, and page 13 line 11 to page 14 line 14.

Reviewer 2 Report

In their manuscript “Differential clinical impact of cachexia and sarcopenia on the prognosis of advanced pancreatic cancer.” Hou et al intended to indicate the clinical distinctions between CC and SC, which can be used to forecast the prognosis of advanced PC patients and to practice personalized care.

This constitutes a large, comprehensive, and broadly coherent body of work that is appreciable.

However, there are a number of minor concerns that can be resolved to improve the quality of the manuscript, listed below.

1. The authors mentioned that the authors checked the levels of hemoglobin and albumin in pancreatic cancer patients, but they did not explain their detection procedure method in the method section.

2. It would be helpful if the authors could explain why low hemoglobin levels are found in CC and SC in pancreatic cancer.

3. The projection of table 1 is a little confusing. Please refer to the numbers in the brackets.

4. Please explain the significance of the CA19-9 level in the discussion section.

5. P-values are missing in numerous locations in Table 3.

6. Any insights from the authors to better understand the recognition of the initial prognosis of CC and OS in pancreatic cancer development.

Author Response

Response to Reviewer 2 Comments

In their manuscript “Differential clinical impact of cachexia and sarcopenia on the prognosis of advanced pancreatic cancer.” Hou et al intended to indicate the clinical distinctions between CC and SC, which can be used to forecast the prognosis of advanced PC patients and to practice personalized care.

This constitutes a large, comprehensive, and broadly coherent body of work that is appreciable.

However, there are a number of minor concerns that can be resolved to improve the quality of the manuscript, listed below.

Reply to Reviewer 2

We thank the reviewer for his/her comments, which have helped to improve the paper and clarify the details of our study. Our point-by-point response is given below (labeled in red).

  1. The authors mentioned that the authors checked the levels of hemoglobin and albumin in pancreatic cancer patients, but they did not explain their detection procedure method in the method section.

Response 1:

Thank you very much for your comments. The description of detection procedure of hemoglobin and albumin in blood samples was added in the Materials and Methods section. Please see the page 5 lines 17-19.

  1. It would be helpful if the authors could explain why low hemoglobin levels are found in CC and SC in pancreatic cancer.

Response 2:

Thank you very much for your comments. The cause of low hemoglobin and albumin levels found in cancer patients with CC and SC was added in the Discussion section. Please see page 23 lines 20-26.

  1. The projection of table 1 (now Table 2) is a little confusing. Please refer to the numbers in the brackets.

Response 3:

Thank you very much for your suggestion. Changes made in Table 1 (now Table 2). Please see page 14 line 16.

  1. Please explain the significance of the CA19-9 level in the discussion section.

Response 4:

Thank you very much for your comments. The significance of the CA19-9 level was added in the Discussion section. Please see page 23 lines 14-19.

  1. P-values are missing in numerous locations in Table 3.

Response 5:

Thank you very much for your comments. The clinical factors affecting overall survival and progression-free survival in advanced pancreatic cancer patients were listed in Table 3 (now Table 1). The bold value indicates that the corresponding parameter had a statistical significance (P < 0.05). Please see page 8 line 14.

  1. Any insights from the authors to better understand the recognition of the initial prognosis of CC and OS in pancreatic cancer development.

Response 6:

Thank you very much for your comments. In this study, we identified the differences in prognostic significance, clinical characteristics, and relevant factors for CC and SC in advanced PC patients. CC represents an independent unfavorable prognostic factor of OS and PFS for all patients and subgroups, whereas SC can be considered as a negative prognostic factor for OS in advanced PC patients within chemotherapy or high BMI subgroups. CC or SC-related factors including older age, female gender and low levels of BMI, hemoglobin, and albumin at diagnosis partially overlap and are interrelated. These data suggest that the differences in wasting pattern or body composition are driven by host or tumor heterogeneity, highlight the need to depict patients' personalized characteristics, and warrant further investigation to early recognition and treatment of the nutritional and metabolic alterations occurring during disease progression and slow the wasting process. Please see page 24 lines 11-13.

Round 2

Reviewer 1 Report

I have now found the manuscript is acceptable for publication. 
Thank you for giving me the opportunity to review this interesting work.